# CORRECTION NETWORKS: META-LEARNING FOR ZERO-SHOT LEARNING

## ABSTRACT

We propose a model that learns to perform zero-shot classification using a meta-learner that is trained to produce a correction to the output of a previously trained learner. The model consists of two modules: a task module that supplies an initial prediction, and a correction module that updates the initial prediction. The task module is the learner and the correction module is the meta-learner. The correction module is trained in an episodic approach whereby many different task modules are trained on various subsets of the total training data, with the rest being used as unseen data for the correction module. The correction module takes as input a representation of the task module's training data so that the predicted correction is a function of the task module's training data. The correction module is trained to update the task module's prediction to be closer to the target value. This approach leads to state-of-the-art performance for zero-shot classification on natural language class descriptions on the CUB and NAB datasets.

## 1 INTRODUCTION

The ability to solve a task without receiving training examples – zero-shot learning – is desirable. This ability is particularly valuable when training data is limited, difficult to obtain, and/or expensive. We as humans can learn new tasks from descriptions of the tasks, as we learn from reading encyclopedia entries, manuals, handbooks, textbooks, etc. Our artificial agents should be able to do the same. However, generalizing to novel tasks unseen during training is challenging. The agent must integrate its previous training experience to solve the new task, without individual training samples from the new task, while avoiding over-fitting on its previous training experience.

We propose a model that learns a correction on predictions in the zero-shot setting, based on the training data set used to generate the initial prediction. Hence, our model is called Correction Networks. The meta-learner corrects for the zero-shot predictions of the learner.

The intuition for our model is that a zero-shot query sample that is different from samples in the training data will require a different correction than a zero-shot query sample that is similar to samples in the training data. Consider a simple classification model trained to identify different color classes. The training dataset for this simple classification problem consists of the classes of {yellow, green, indigo}. Let there be two zero-shot classes: pink and blue. The deviation of predicted values from true values of pink are different, possibly larger, than deviations of predicted values from true values of blue. These deviations would again be different if the training dataset consisted of the classes of {white, red, purple}.

Correction Networks update the predictions based on the training data. This updated prediction is trained to be closer to the target value than the original prediction. Correction Networks consist of two modules: a *task module* that supplies an initial prediction, and a *correction module* that provides a correction to the initial prediction. The task module is the learner and the correction module is the meta-learner. The final prediction is the task module's initial prediction combined with the correction module's correction. This method is illustrated in Figure 1.

The prediction of the meta-learner is used to modify the output of the learner, unlike other meta-learning algorithms that learn to update, initialize, or optimize the learner's weights (Finn et al., 2017) (Ravi & Larochelle, 2017). Our approach is for zero-shot learning while previous meta-learning approaches focus on few shot learning and require a few training samples for the new task.

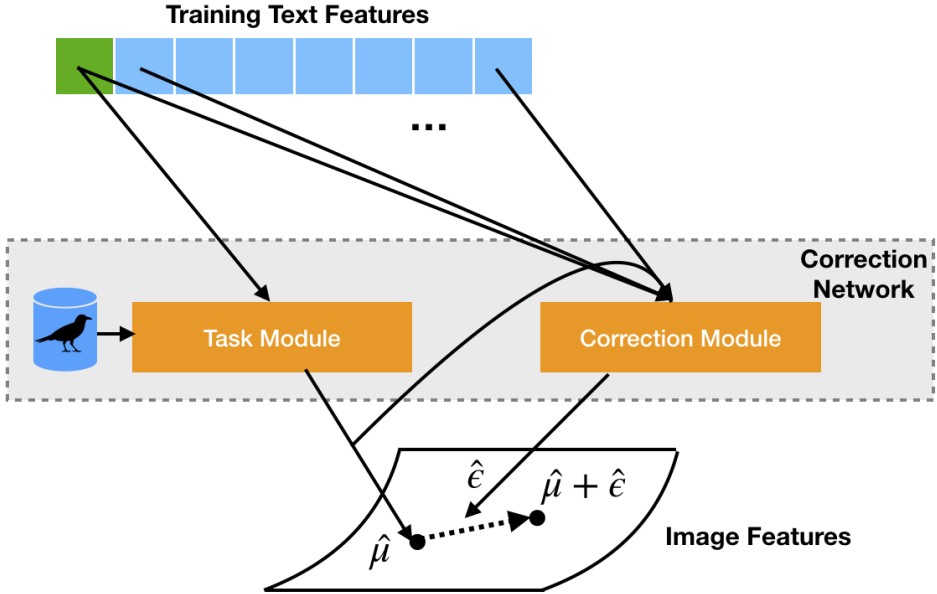

Figure 1: Correction Networks consist of two modules: a task module and a correction module. The task module produces an initial prediction. The correction module provides a correction such that when the correction is combined with the initial prediction, the resulting prediction is better than the initial prediction.

Another novelty is that the correction module (the meta-learner) takes as input the dataset used to train the task module.

Correction Networks can be used for problems for which an updated prediction can be obtained by applying a correction. Correction Networks are independent of the representation of the task module. Existing models that provide predictions can be treated as task modules. The correction module only need the inputs to the task module, the predictions of the task module, and the training data of the task module. While our focus is on neural network architectures, both the task module and correction modules can be, but needs not be, neural networks.

Training proceeds by randomly partitioning the training data $\mathcal{D}$ to create disjoint sets $\mathcal{D}_S$ and $\mathcal{D}_U$. In the case of zero-shot classification, the partition is done by class so that the classes in $\mathcal{D}_S$ and $\mathcal{D}_U$ are disjoint. The task module is trained on $\mathcal{D}_S$. After the task module is trained, $\mathcal{D}_U$ is used as the 'zero-shot' queries for the task module. The task module supplies initial predictions on $\mathcal{D}_U$. The correction module is trained to predict the corrections that need to be applied to the task module's initial predictions to obtain the true values associated with $\mathcal{D}_U$. One novelty is that the correction module, the meta-learner, also takes as input the dataset used to train the task module. The prediction of the meta-learner is used to modify the output of the learner, unlike other meta-learning algorithms that learn to update or initialize the learner's weights (Finn et al., 2017) (Ravi & Larochelle, 2017).

We demonstrate Correction Networks on the problem of zero-shot image classification based on natural language class descriptions. Zero-shot image classification involves classifying images into classes that were unseen in the training data, given a natural language description of the new class.

The main contribution of this paper is a zero-shot learning model that corrects zero-shot predictions based on training data. We demonstrate our model on zero-shot image classification with class text descriptions. Our evaluation shows that Correction Networks compares favorably to state-of-the-art zero-shot learning methods.

The rest of this paper is as follows: Section 2 covers related work, Section 3 describes our Correction Network model, Section 4 summarizes the experiments and results, and Section 5 finishes with conclusions.

## 2   RELATED WORKS

For zero-shot learning for images, the majority of state-of-the-art methods are embedding-based methods. This often involves learning a mapping from the visual space to the semantic space of class labels or vice versa. Alternatively, the embedding function between the visual and semantic spaces is jointly learned through a latent space.

To represent the new, novel classes, hand crafted attributes are often used. The idea is that given a defined attribute ontology, each class name can be converted to an attribute vector against which image features are compared (Lampert et al., 2009) (Palatucci et al., 2009) (Farhadi et al., 2009) (Akata et al., 2013). Attribute based embedding spaces more recently gave way to text based embeddings. These text-based embeddings learn from a large external text corpus e.g. Wikipedia or WordNet on a natural language task to learn word embeddings into which class names are then projected (Oquab et al., 2014) (Socher et al., 2013) (Frome et al., 2013) (Fu et al., 2014).

Semantic representations for zero-shot classes have also been created from text documents of the classes eg. Wikipedia articles corresponding to the class. This includes learning attribute representations from a web source by ranking the visualness of attribute candidates (Berg et al., 2010). Internet sources have also been mined for semantic relatedness of attribute-class association (Rohrbach et al., 2013). Directly measuring the compatibility between documents describing the class and visual features have also been modeled using deep learning (Zhu et al., 2018) (Ba et al., 2015) and non-deep learning models (Elhoseiny et al., 2013) (Romera-Paredes & Torr, 2015) (Fu et al., 2015) (Qiao et al., 2016) (Elhoseiny et al., 2017a).

A more recent strategy frames zero-shot recognition as a conventional supervised classification problem by hallucinating samples for unseen classes. Classification performance thus depends on the quality of the hallucinated samples. Guo et al. (2017a) assumes the visual features of each class is distributed as a Gaussian and estimates the distribution of unseen classes by linearly combining the distributions of seen classes. Long et al. (2017) synthesizes visual data from attributes of classes in a one-to-one mapping. Guo et al. (2017b) assigns pseudo labels to samples from seen classes. Zhu et al. (2018) uses generative adversarial training with the generator learning to generate image features from text and then query samples are classified based on k-nearest neighbors to labeled or generated samples. In contrast, our approach avoids the data generation phase and instead, directly predicts the center of classes in image feature space. Classification regions are thus defined based on the class of the nearest class center.

Our work is similar to gradient boosting which produces a prediction model using an ensemble of weak models, where each additional model adds an estimator to provide a better model. Unlike gradient boosting, our approach takes in the training data used to create the initial estimate so that the model learns how the initial estimate deviates due to the training data. While the initial learner in gradient boosting is a weak learner, we use a strong learner.

## 3   CORRECTION NETWORKS

### 3.1   NOTATION

Let $\mathcal{D}_S$ denote our training data and $\mathcal{D}_U$ our testing data. $\mathcal{D}_S$ is subdivided into disjoint sets $\mathcal{D}_S^s$ and $\mathcal{D}_S^u$. For classification, the classes in $\mathcal{D}_S^s$ are disjoint from the classes in $\mathcal{D}_S^u$.

### 3.2   MODEL

Correction Networks $M$ consists of two modules: a task module $M_T$ and a correction module $M_C$. The model components and their inputs and outputs are illustrated in Figure 2.

The task module $M_T$ is so-called because it is task specific and related to the application. The task module is trained on $\mathcal{D}_S^s$. The output of $M_T$ is an estimate $\hat{\mu}$ of a target $\mu$. The predictions of the task module on its training data $\mathcal{D}_S^s$ is $\hat{\mu}_S^s$. Training the task module proceeds by minimizing the distance between $\hat{\mu}_S^s$ and the ground truth $\mu_S^s$. A sparsity regularization term is added to the input layer of $M_T$. The loss is:

$$L_{M_T} = \mathbb{E}[d(M_T(T_S^s), \mu_S^s)] + \alpha ||w_{M_T, input}||^2 \qquad (1)$$

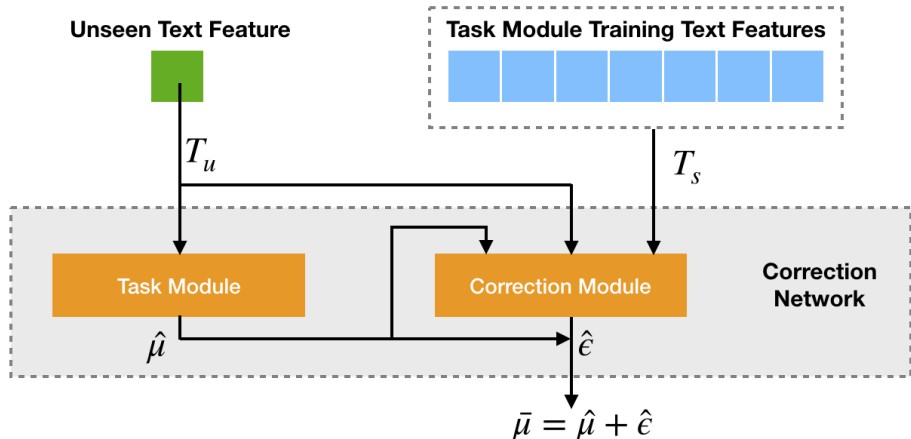

Figure 2: Inputs and outputs of Correction Networks demonstrated on zero-shot image classification based on a natural language description of the zero-shot class. The task module takes as input the natural language description of the zero-shot class and makes an initial prediction of the cluster center of the class in image feature space. The correction module improves this initial prediction by applying a correction, taking as input the task module's initial prediction, training data, and the zero-shot class description.

where $T$ is the class text description, $\mu$ is the empirical mean of samples that belong to the class, and $d$ is a distance function. We use the L2 norm as the distance function.

The task module $M_T$ is not trained on $\mathcal{D}_S^u$ nor $\mathcal{D}_U$. The task module's predictions on $\mathcal{D}_S^u$ are $\hat{\mu}_S^u$. Likewise, the task module's predictions on $\mathcal{D}_U$ are $\hat{\mu}_U$. The correction module $M_C$ computes a correction $\hat{\epsilon}_S^u$ that is applied to the prediction $\hat{\mu}_S^u$ of the task module $M_T$, where $\hat{\epsilon}_S^u$ is calculated based on the data used to train $M_T$, such that the corrected prediction $(\hat{\mu}_S^u + \hat{\epsilon}_S^u)$ is closer than $\hat{\mu}_S^u$ to the ground truth $\mu_S^u$. Training the correction module proceeds by minimizing the distance between $\mu_S^u$ and $(\hat{\mu}_S^u + \hat{\epsilon}_S^u)$. We use the L2 norm. The training data for the task module $D_S^s$ is input into the correction module by representing the training data $D_S^s$ as an un-ordered collection of data by using a pooling function. We use linear layers followed by sum pooling. The objective function of the correction module is to minimize:

$$L_{M_C} = \mathbb{E}[d(M_C(T_S^u), \mu_S^u - M_T(T_S^u))] \tag{2}$$

We adopt the meta-learning sampling strategy for training as in Snell et al. (2017). Training data for Correction Networks is formed by randomly selecting a subset $\mathcal{D}_S^s$ from the training data $\mathcal{D}_S$. Then, the task module $M_T$, is trained on $\mathcal{D}_S^s$. The remaining tasks that the task module $M_T$ does not train on are treated as $\mathcal{D}_S^u$ for $M_T$. The algorithm is detailed in Algorithm 1.

To use Correction Networks for evaluation, the task module $M_T$ outputs $\hat{\mu}_U$ and the correction network supplies $\hat{\epsilon}_U$. The output of the Correction Networks is $\bar{\mu}_U = \hat{\mu}_U + \hat{\epsilon}_U$. The evaluation algorithm is outlined in Algorithm 2.

Correction Networks can be used for zero-shot problems for which an updated prediction can be obtained by applying a correction. Appropriate problems are those with continuous valued outputs. For classification, samples can be represented as continuous values in a feature space and then classified based on the nearest class center. In our experiments, both the base network and the Correction Network are deep neural networks.

### 3.3 CORRECTION NETWORKS FOR ZERO-SHOT CLASSIFICATION

In zero-shot classification, there are seen classes $S$ available during training and unseen classes $U$ during evaluation, where $S$ and $U$ are disjoint.

We are given data $\mathcal{D}_S = \{(x_n, y_n), t_m\})$ containing samples $\{x_n, n = 1, ..., N\} \in X$ and class labels $\{y_n, i = 1, ..., N, y_n \in Y\}$ such that $y_n \in S$, along with class text descriptions $\{t_m, m =$

$1, ..., M\} \in T_S$ where there exists exactly one $t_m$ for each class in $S$. For image classification, each sample $x_n$ is an image's features extracted from a pre-trained model. There is no textual data per image. The only text data is $t_m$ with one text description per class.

At evaluation, we are given new class descriptions for classes in $U$, denoted by $T_U$. We seek to learn a function $f$ to minimize the 0-1 loss between the predicted $f(x_i)$ and true class labels $y_i$ for each sample $x_i$ where the samples and class labels can be from the new classes in $U$.

Correction Networks will map the class text descriptions $T_U$ to their corresponding class centers in image feature space $\hat{\mu}_U$. Classification of a single image is done by assigning it the class label of the closest class center, measured by L2 distance. We subdivide the training classes $S$ into $S^s$ and $S^u$. Then, $\mathcal{D}_S^s$ corresponds to the data from $\mathcal{D}_S$ that belong to the classes in $S^s$. Similarly, $\mathcal{D}_S^u$ corresponds to the data from $\mathcal{D}_S$ that belong to the classes in $S^u$. Training and evaluation then proceed as described in the previous section. Correction Networks predict class cluster centers for zero-shot classes. Individual samples are classified to the nearest class cluster center using L2 distance. For classes in the training data, the class cluster center is the empirical mean of samples from the respective class.

In terms of implementation, both the task module and the correction module are feed-forward neural networks consisting of linear layers, activation functions, and dropout. We found the task module performance improves slightly when the output of the task module is fed into a classifier with a single hidden layer that is also trained to classify samples from the task model's training dataset. Additional model and training details are in Appendix A.

---

**Algorithm 1** Correction Networks training algorithm. $K$ is the number of classes in the training set $S$. $S$ is partitioned into $S^s$ and $S^u$ by class. $T_S$ denotes the class descriptions of classes in $S$. $\mathcal{D}_S$ is the set of individual samples of $(x_i y_i)$ for the classes in the training set $S$. $\mathcal{D}_S^s$ denotes the subset of $\mathcal{D}_S$ containing all samples from classes in $S^s$. $\mathcal{D}_k$ is the subset of $\mathcal{D}_S$ containing the individual elements $(x_i y_i)$ such that $y_i = k$. MEAN is a function that returns the mean of its inputs. OPTIMIZER is an optimizer. $d$ is a distance function, for example, L2 distance. $M_T$ is the task module. $M_C$ is the correction module.

1: **for** $k$ in 1 to $K$ **do**
2:     $\mu_k \leftarrow \text{MEAN}(\mathcal{D}_k)$
3: **end for**
4: **while** not done **do**
5:     $S^s \leftarrow \text{RANDOMSAMPLE}(\{1, ..., K\}, s)$
6:     $S^u \leftarrow \{1, ..., K\} \backslash S^s$
7:     **while** $M_T$ has not finished training **do**
8:         $\hat{\mu}_S^s \leftarrow M_T(T_S^s)$
9:         $J_{M_T} \leftarrow d(\hat{\mu}_S^s, \mu_S^s) + \alpha||w_{M_T}||^2$
10:         $w_{M_T} \leftarrow \text{OPTIMIZER}(\nabla_{w_{M_T}} J_{M_T}, w_{M_T})$
11:     **end while**
12:     $\hat{\mu}_S^u \leftarrow M_T(T_S^u)$
13:     $\hat{\epsilon}_S^u \leftarrow M_C(T_S^u, \hat{\mu}_S^u, T_S^s)$
14:     $\epsilon_S^u \leftarrow \mu_S^u - \hat{\mu}_S^u$
15:     $J_{M_C} \leftarrow d(\hat{\epsilon}_S^u, \epsilon_S^u)$
16:     $w_{M_C} \leftarrow \text{OPTIMIZER}(\nabla_{w_{M_C}} J_{M_C}, w_{M_C})$
17: **end while**

---

**Algorithm 2** Correction Networks evaluation algorithm. $S$ is the set of classes in the training set. $U$ is the set of zero-shot classes. $T_U$ denotes the class descriptions of zero-shot classes in $U$.

**Require** Trained $M_T$ on $T_S^s$, trained $M_C$
**Output:** Prediction $\bar{\mu}_U$
1: $\hat{\mu}_U \leftarrow M_T(T_U)$
2: $\hat{\epsilon} \leftarrow M_C(T_U, \hat{\mu}_U, T_S^s)$
3: $\bar{\mu}_U = \hat{\mu}_U + \hat{\epsilon}_U$

---

**Natural language description of a class**      **Samples from the class**

| |
|---|
| The Mangrove Cuckoo...Adults have a long tail, brown above and black-and-white below, and a black curved bill with yellow on the lower mandible  The head and upper parts are brown  There is a yellow ring around the eye  This bird is best distinguished by its black facial mask and buffy underparts This cuckoo is found primarily in mangrove swamps... |
| The Common Yellowthroat... are small songbirds that have olive backs, wings and tails, yellow throats and chests, and white bellies. Adult males have black face masks which stretch from the sides of the neck across the eyes and forehead, which are bordered above with white or gray. Females are similar in appearance, but have paler underparts and lack the black... |
| The Green-tailed Towhee, Pipilo chlorurus, is the smallest towhee, but is still one of the larger members of the "American sparrow" family ... This bird can be recognized by the bright green stripes on the edge of its wings. It has a distinct white throat and a rufous cap. It is fairly tame, but often stays hidden under a bush. It is uncommonly seen because of its... |

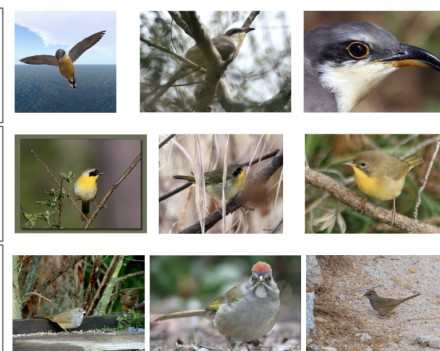

Figure 3: Examples from the dataset where there is one natural language description of a class and the goal is to classify images for new classes without training sample images for the new classes.

# 4 EXPERIMENTS

## 4.1 TASK SETUP

We demonstrate Correction Networks on fine-grained zero-shot classification based on natural language text descriptions of the class. Fine-grained image classification is image classification when classes are very similar. This requires the ability to distinguish between minute details and subtle differences between classes. An example of fine-grained classification is distinguishing between different birds that may be very similar.

Features of a class described in a text description may not be visible in all images belonging to that class. This loss of visibility can be due to image cropping, posture, camera angle, age, and gender of the bird, among other reasons. For example, an image shown from the front results in obfuscation of the back. An image may only be a head shot. Other objects such as branches or leaves may also block the object of interest. This is illustrated in Figure 5.

## 4.2 DATASETS

**Image Data** We evaluate our method on Caltech UCSD Birds 2011 (CUB) (Wah et al., 2011) and North America Birds (NAB) (Van Horn et al., 2015). The CUB dataset contains 200 classes of birds with a total of 11,788 images. We only use the images and their associated labels. We use a newer version of the NAB dataset that contains 404 classes with a total of 48,562 images. Two splits were proposed for each dataset (Elhoseiny et al., 2017b). The splits are named *Super-Category-Shared* (SCS) and *Super-Category-Exclusive* (SCE). In the SCS split, unseen classes and seen classes share the same parent category. In the SCE split, the parent categories are disjoint between seen and unseen classes. The SCE splits are more difficult than the SCS splits because the unseen classes are more different than the seen classes. Where published validation sets are not available, we create our own validation sets by holding out part of the training set, where the classes in the validation set are randomly sampled from among the training classes.

**Visual Features** The visual features are activations extracted from the FC layer of a pre-trained detector network (Elhoseiny et al., 2017b). We use the published features from (Zhu et al., 2018).

**Text Features** Each class is associated with a Wikipedia article. The articles are tokenized into words, stop words are removed, and words are reduced to their word stems. Then, the processed text is represented by TF-IDF features. We use the published text features from (Zhu et al., 2018). Examples of the text and images are shown in Figure 3.

Table 1: Zero-shot learning classification results accuracy @ 1 on the CUB-200-2011 dataset and the NAB dataset using class descriptions from Wikipedia on the Super-Category-Shared (SCS) and Super-Category-Exclusive (SCE) zero-shot splits

| METHOD | CUB | | NAB | |
|---|---|---|---|---|
| | SCS | SCE | SCS | SCE |
| MCZSL (Akata et al., 2016) | 34.7 | - | - | - |
| WAC-Linear (Elhoseiny et al., 2013) | 27.0 | 5.0 | - | - |
| WAC-Kernel (Elhoseiny et al., 2017a) | 33.5 | 7.7 | 11.4 | 6.0 |
| ESZSL (Romera-Paredes & Torr, 2015) | 28.5 | 7.4 | 24.3 | 6.3 |
| SJE (Akata et al., 2015) | 29.9 | - | - | - |
| ZSLNS (Qiao et al., 2016) | 29.1 | 7.3 | 24.5 | 6.8 |
| $SynC_{fast}$ (Changpinyo et al., 2016) | 28.0 | 8.6 | 18.4 | 3.8 |
| $SynC_{OVO}$ (Changpinyo et al., 2016) | 12.5 | 5.9 | - | - |
| ZSLPP (Elhoseiny et al., 2017b) | 37.2 | 9.7 | 30.3 | 8.1 |
| GAZSL (Zhu et al., 2018) | 43.7 | 10.3 | 35.6 | 8.6 |
| Correction Networks | **45.8** | 10.0 | **37.4** | **9.5** |

Table 2: Generalized Zero-shot learning classification area under Seen-Unseen Curve on CUB

| METHOD | CUB | | NAB | |
|---|---|---|---|---|
| | SCS | SCE | SCS | SCE |
| WAC-Linear (Elhoseiny et al., 2013) | 23.9 | 4.9 | 23.5 | - |
| WAC-Kernel (Elhoseiny et al., 2017a) | 22.5 | 5.4 | 0.7 | 2.3 |
| $SynC_{Fast}$ (Changpinyo et al., 2016) | 13.1 | 4.0 | 2.7 | 0.8 |
| ESZSL (Romera-Paredes & Torr, 2015) | 18.5 | 4.5 | 9.2 | 2.9 |
| ZSLNS (Qiao et al., 2016) | 14.7 | 4.4 | 9.3 | 2.3 |
| $SynC_{OvO}$ (Changpinyo et al., 2016) | 1.7 | 1.0 | 0.1 | - |
| ZSLPP (Elhoseiny et al., 2017b) | 30.4 | 6.1 | 12.6 | 3.5 |
| GAZSL (Zhu et al., 2018) | 35.4 | 8.7 | 20.4 | 5.8 |
| CorrectionNet | **41.9** | **9.0** | **25.4** | **7.6** |

## 4.3 Conventional Zero-Shot Recognition

The top-1 accuracy of our method and eight state-of-the-art algorithms for the CUB and NAB datasets for both the SCS split and the SCE split are tabulated in Table 1. The eight comparison models are MCZSL (Akata et al., 2016), ZSLNS (Qiao et al., 2016), SJE (Akata et al., 2015), WAC (Elhoseiny et al., 2017a), SynC (Changpinyo et al., 2016), ZSLPP (Elhoseiny et al., 2017b), and GAZSL (Zhu et al., 2018). The performance numbers are copied from (Zhu et al., 2018). MCZSL directly uses manual part annotations to extract visual representations. On the other hand, our approach, GAZSL, and ZSLPP merely use detected parts for both training and testing. Thus, methods that use detected parts are expected to perform poorer than MCZSL, which uses manual parts annotations. Our model performs favorably against the other models, showing a relative improvement of up to 10% over the previous state-of-the-art. Qualitative results are shown in Figure 4 in Appendix B.

## 4.4 Generalized Zero-Shot Learning

The conventional zero-shot learning setting considers queries that come from unseen classes $\mathcal{D}_U$ and classification of queries is restricted to be among the unseen classes $U$. In contrast, the generalized zero-shot learning setting classifies queries from both seen $\mathcal{D}_S$ and unseen classes $\mathcal{D}_U$ into $S \cup U$. A metric for generalized zero-shot learning performance is the area under the seen-unseen curve (Chao et al., 2016). This is tabulated in Table 2 with values from competitors copied from Zhu et al. (2018). For Correction Networks, Correction Networks is used to predict $\hat{\mu}$ for the unseen classes

Table 3: Effects of different components on zero-shot classification accuracy (%) on CUB SCS

| METHOD | CUB |
|---|---|
| Task module only | 43.8 |
| Correction module without task module's training dataset | 43.4 |
| Correction Networks | 45.8 |

while the empirical $\mu$ is used for the seen classes. Correction Networks consistently outperforms previous approaches in the generalized zero-shot learning setting. This improvement is between 3-31% relative to the runner-up.

## 4.5 ABLATION STUDIES

To examine the contributions of separate components of our model, we conduct ablation studies by removing selected components. Then, the model is retrained and evaluated on the test set. The resulting performance of the ablated models are reported in Table 3.

Removing the correction module degrades the performance, but the task module itself already achieves the state of the art. The correction module improves the performance by 2% in absolute accuracy. This suggests that the correction modules makes a slight change to the output of the task module. When we remove the task module's training data as input into the correction module, the zero-shot accuracy decreases by 2.4%. This demonstrates that the task module's training data is important to the correction module's prediction.

## 4.6 EXTENSIONS

Correction Networks are independent of the architectures of the modules and can be used for problems for which an updated prediction can be obtained by applying a correction. Additional investigations include applying Correction Networks to other tasks with outputs that can be updated, including for example, estimates for regression problems, classification probabilities, or probability distributions for reinforcement learning policies. The type of update can also be multiplicative, weighted sum, or another function. In addition to the zero-shot classification problem presented here, Correction Networks can also be applied to correct outputs in the few shot setting.

The magnitude of the correction vector predicted by the Correction Network can be interpreted as the size of the update applied. It would be interesting to use an interpretable output space where different dimensions correspond to human-understandable variables. Then, the size and direction of corrections in individual dimensions can be interpreted. The size of the correction can be compared against the number of training samples given to the base network, defined as the number of seen classes. One would expect that the size of the correction decreases as the number of seen classes increases. Also, one would expect that the accuracy stays constant or else improves with the number of seen classes. It would be interesting to see if correction networks can be used to understand bias in the training data.

## 5 CONCLUSION

We propose a zero-shot learning model that consists of a task module and a correction module that is trained to extrapolate from training data to unseen data. The training data is partitioned into a set of data used to train the task module and a disjoint set of data used to train the correction module. This later data is zero-shot with respect to the task module and the correction module is trained to update the task module's zero-shot predictions to create a better prediction.

Our model is demonstrated on zero-shot fine-grained classification using only a single natural language description per zero-shot class. Our model performs favorably against the state-of-the-art on zero-shot classification. This framework is flexible to different representations of the task and correction modules and can be extended to other types of model predictions that can be updated to improve predictions.

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

APPENDIX A: EXPERIMENTAL DETAILS

**Architecture**

The task module is a feed-forward network consisting of linear layers. The input is the TF-IDF feature for a class. There are two outputs: the predicted mean in image embedding space and softmaxes over the classes in $T_S$. The loss function is the sum of the L2 loss between $\hat{\mu}_S^u$ and $\mu_S^u$ and the classification loss, with a sparsity regualarizer on the input layer.

The correction module is also a feed-forward network consisting of linear layers. Each sample in $T_S^s$ is fed through linear layers, then sum pooled across classes, and fed through an additional layer. The final transformation of $T_S^s$ is concatenated with $T_S^u$ and $\hat{\mu}_S^u$ and then passed through linear layers to output $\hat{\epsilon}_S^u$. The loss function is L2 loss between $\hat{\epsilon}_S^u$ and $\epsilon_S^u$. Architectures are illustrated in Figure 4.

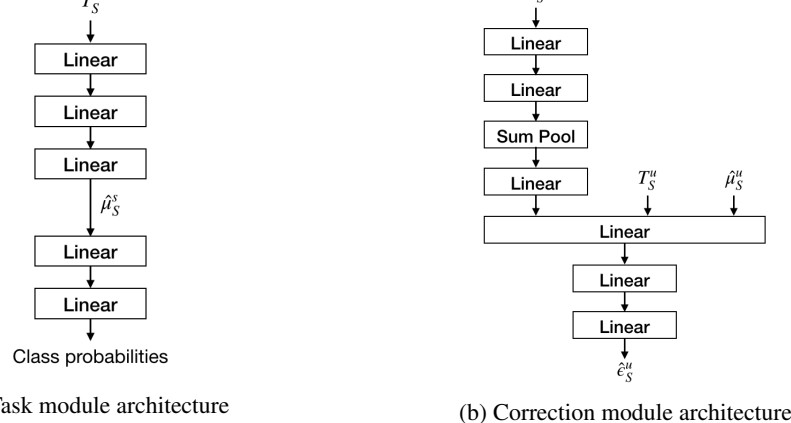

(a) Task module architecture

(b) Correction module architecture

Figure 4: Architecture of the modules

**Training**

The publicly available SCS and SCE splits (Elhoseiny et al., 2017b) define a test set. The remaining classes are randomly divided into a validation set and a training set. The size of the validation set is the same size as the training set. For example, CUB has 200 classes with the conventional SCS split consisting of 50 test classes. We divided the remaining 150 classes into a 50 class validation set and a 100 class training set.

The Correction Network is first trained by considering the training set as $T_S$. Training proceeds as in Algorithm 1. That is, $T_S$ is randomly divided into disjoint sets $T_S^s$ and $T_S^u$. A task module is trained on $T_S^s$, stopping based on performance on the validation set. The trained task module predicts $\hat{\mu}_S^u$ based on $T_S^u$. The correction module is updated using $T_S^u$, $\hat{\mu}_S^u$, and $T_S^s$. The correction module is trained across multiple episodes of task module training, where the task module in each episode is trained on a randomly sampled $T_S^s$ from $T_S$. Training of different episodes of task modules is parallelizable as they are independent of one another and independent of the correction module.

Training of the correction module stops based on performance on the validation set. The stopping iterations of the task modules and correction module on the validation set is used to determine the stopping iterations of the task modules and the correction module on the test set.

To evaluate on the test set, the validation and training sets are combined and treated as $T_S$ while the test set becomes $T_U$. Training again proceeds as in Algorithm 1. The Adam optimizer is used to train the models.

APPENDIX B: RESULTS VISUALIZATIONS

Correctly and incorrectly classified images are shown in Figure 4 below. Each row is a different class. The left-most three images are correctly classified. The middle three images are false negatives, and the right-most images are false positives.

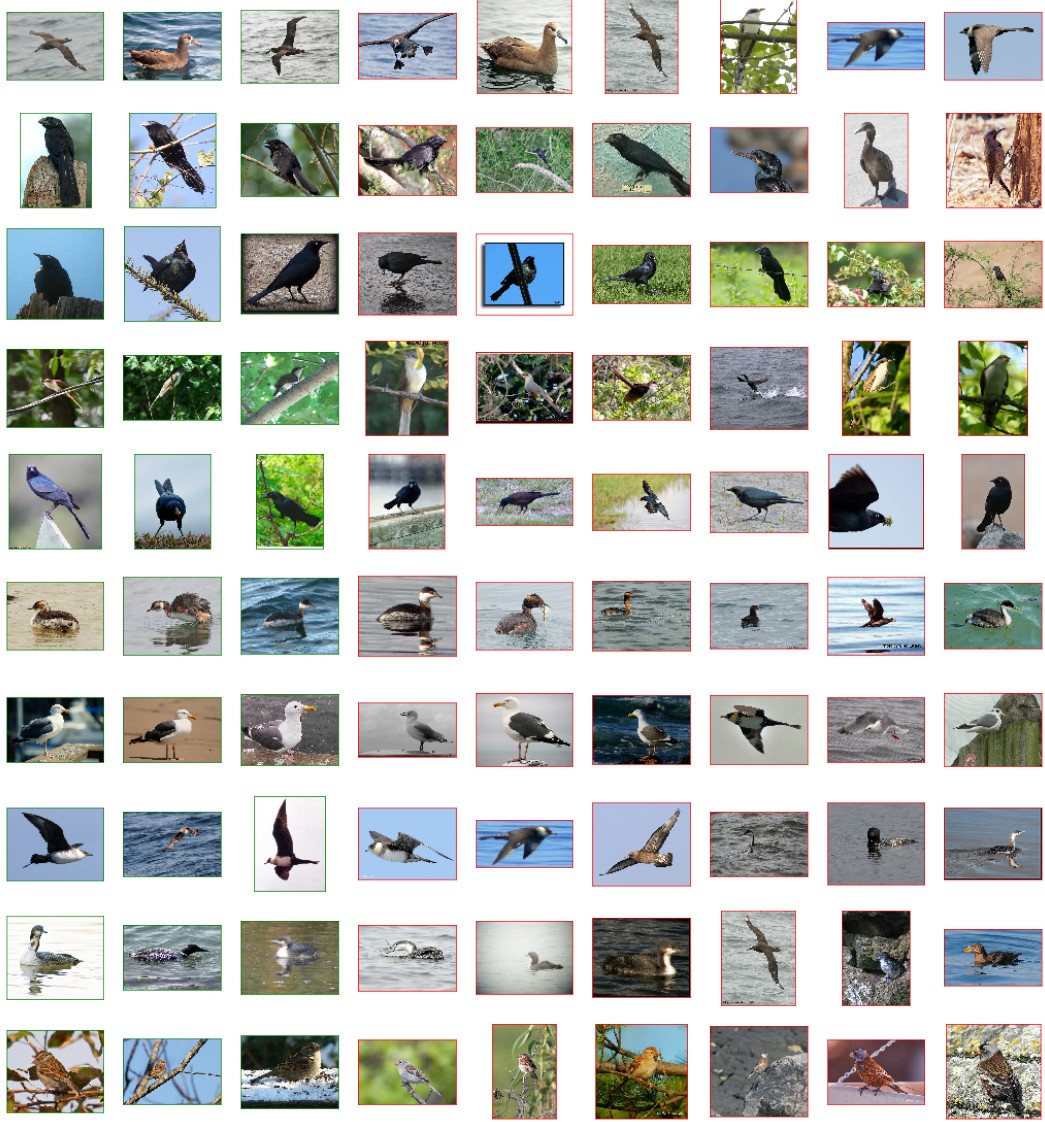

Figure 5: Qualitative classification results. Each row is a different class with correctly classified examples (left three), false negatives (middle three), and false positives (right three).

