# OpenReview forum: "Correction Networks: Meta-Learning for Zero-Shot Learning"
_ICLR.cc/2019/Conference_

### Official Review · AnonReviewer1 · 2018-10-31
**Original approach with strong results, but lacks many details**

**Rating:** 7
**Confidence:** 4

**Review:**

=== Post-rebuttal update ===

The authors' rebuttal provided many of the details I was seeking. I asked a few additional questions which were also recently addressed, and I encourage the authors to include these clarifications into the final draft of the paper.

Hence, I've increased my score for this paper.

=== Pre-rebuttal review ===
This paper presents a meta-learning approach to zero-shot learning. The idea is to train a correction module which is trained to produce a correction to the output of a previously trained task module. The hypothesis is that the correction should depend on the nature of the training data of the task module, and so the correction module receives as input a representation of the training data of the task module. An episodic approach is then used for training the correction module, whereby many different task modules are trained on various subsets of the total training data, the rest being used as unseen data for the correction module.

The proposed idea is original and the results are strong. Generally, I'd be inclined to see this paper published.

However right now, the paper lacks A LOT of details on how the experiments were run. I would like to see these answered in the rebuttal, before I consider raising my rating for this paper:
- What are the architectures used for M_T and M_C?
- What distance functions was used for training?
- What optimizer was used for training?
- How was convergence established in the inner and outer while loops of algorithm 1?
- Text mentions that before evaluation, M_T is trained on all data in D_S. How is this done exactly (e.g. how is convergence assessed)?
- How is the T_S computed exactly?
- How expensive is it to run Algorithm 1 (i.e. to train the correction module)? Since a new task module M_T needs to be trained for each subset S^s, it seems like it might be expensive to run... if not, why?

I would also strongly suggest the authors release their code if this paper ends up being published.

In summary:

Pros
- Claims SOTA results on two good benchmarks for zero-shot learning
- Approach is original

Cons
- Paper lacks a lot of methodological and experimental details

Some minor details:

- "We found the task module performance improves slightly when the output of the task module is feed into a classifier with a single hidden layer that is also trained to classify samples from the task model’s training dataset." => I don't understand what this means. Isn't the output of the task module already trained to classify samples from its training dataset? So why is this additional single hidden layer needed?
- Typos:
  - on few shot learn => on few shot learning
  - but needs not => but need not
  - image image classification => image classification
  - the the compatibility => the compatibility
  - psuedo => pseudo
  - "The task module is trained to minimize" => that reads like an unfinished sentence
  - \hat{\mu}_U \hat{\mu}_U => \hat{\mu}_U
  - inputted => input
  - FOr => For
  - it's inputs => its inputs
  - otherhand => other hand

---

> ### Author Response · Authors · 2018-11-21
> **Re: Additional details**
>
> We thank the Reviewer for the comments. We answered the reviewer’s questions below and we added this information into our revised draft:
>
> Comment: - What are the architectures used for M_T and M_C?
> The architectures of M_T and M_C are illustrated in the new additional Appendix A. The architectures consist of linear feedforward layers.
>
> Comment: - What distance functions was used for training?
> The L2 distance functions were used for training.
>
> Comment: - What optimizer was used for training?
> The ADAM optimizer was used.
>
> Comment: - How was convergence established in the inner and outer while loops of algorithm 1?
> Convergence of the task module M_T is established by monitoring the loss of M_T on a validation set. The validation set was created by randomly sampling classes from the training set to create a validation set of the same size as the training set. For example, CUB has 200 classes with the conventional SCS split having 50 test classes. We divided the remaining 150 classes into a 50 class validation set and a 100 class “training” set. Training of M_T stops once the loss of M_T on the validation set stops improving. This stopping iteration is also used during testing when there is no validation set. During testing, the validation set is recombined with the “training” set. Similarly, convergence of M_C is established by monitoring the loss of M_C on the validation set.
>
> Comment: - Text mentions that before evaluation, M_T is trained on all data in D_S. How is this done exactly (e.g. how is convergence assessed)?
> Training of M_T stops on a stopping iteration determined from using a validation set. We experimented with using all data in D_S to train a M_T and using this M_T for evaluation along with the trained M_C. However, we have since found that using a M_T used to train M_C resulted in better performance. We believe this is because T_s^s is sum pooled and using T_s as T_s^s results in values that are outside of the training distribution of M_C because sum pooling across T_s results in larger values than sum pooling across T_s^s.
>
> Comment: - How is the T_S computed exactly?
> Each text description is represented as a vector of TF-IDF features. This vector is then passed through linear layers and then sum pooled across classes. The architecture is shown in Appendix A.
>
> Comment: - How expensive is it to run Algorithm 1 (i.e. to train the correction module)? Since a new task module M_T needs to be trained for each subset S^s, it seems like it might be expensive to run... if not, why?
> Each M_T takes about 2-10 minutes to train on a single GPU. Training of episodes of task modules is parallelizable as they are independent of one another and independent of the correction module. The output from the task modules are saved and cached to be re-used for different experiments on the correction module (e.g. hyperparameter tuning of the correction module). The correction module itself takes about an hour to train on a single GPU.
>
> Comment: - We found the task module performance improves slightly when the output of the task module is feed into a classifier with a single hidden layer that is also trained to classify samples from the task model’s training dataset." => I don't understand what this means. Isn't the output of the task module already trained to classify samples from its training dataset? So why is this additional single hidden layer needed?
>
> The output of the task module is used to classify samples from the training dataset using L2 distance between the image sample and the class center as predicted by the task module.  The task module is trained to predict the mean of the image samples in a class, given a text description of the class. This prediction is trained using L2 loss. The additional single hidden layer takes as input the predicted mean of image samples and predicts softmax class probabilities using cross entropy loss. This additional single hidden layer is also trained on image samples to predict softmax class probabilities.
> So the architecture is: linear layers -> predicted class mean -> additional single hidden layer -> softmax. The total loss is the sum of L2 and cross-entropy.  A hypothesis as to why this improves performance is the classifier is more sensitive in certain dimensions (and their combinations) than others, and this loss is combined with the L2 loss which is otherwise dimension invariant. This additional single hidden layer improved the accuracy of the task module by 0.5% to 1% in absolute accuracy.
>
> Thank you for the minor details. We appreciate your attention to detail.

---

### Official Review · AnonReviewer2 · 2018-11-01
**Interesting idea, but many flaws.**

**Rating:** 4
**Confidence:** 4

**Review:**

This paper presents an interesting idea by formulating the problem of zero-shot learning in a meta-learning framework.  Specifically, the proposed model consists of two components: the task module and the correction module, where the former module learns to map the text description of a class to the sample mean and the latter one updates the predictions for unseen classes.

The presentation of this paper is very poor.  Proposed meta-framework has some flaws. And, the experiments are not persuasive enough to demonstrate the significance of the proposed framework.

The proposed zero-shot classifier is based on the nearest centroid.   Authors formulate the learning problem as mapping the text description of each class to sample mean of the data of the class.   Within a meat-training instance, the training performance is based on L2 distance between the mapped mean and the sample mean of each class.   This setup is wired.  This because, no matter how many data (x, y pairs) we get, the proposed method only makes the prediction based on the pre-calculated mean.  In other words, the "number of samples" in a meta-training dataset becomes the number of unique classes appears in training.    For instance, if we have 10 classes in the $D_\mathcal{S}$, and10000 samples per class,  the proposed setup will consider the meta training only consist of 10 data points.

In addition, the proposed method heavily rely on the feature extractor of the image.  The classification performance could be poor if two the mean of different classes close to each other.    Even they are not, the proposed framework cannot provide sample-level generalization.

Another confusion I have is why the training of the task module is not based on a fixed correction module?

The experiments also have many problems.  Authors need to clearly state how they construct meta-training, validation and testing instance.   Since the proposed framework is a meta-framework, authors need to report their performance in different meta-train/test splits.  The conventional split of CUB and NAB is only considered as a single split.  How well the proposed framework generalizes to other meta splits?     How well the proposed method performance to a generalized zero-shot setting?

There are many typos.  Auhtors definitely need to improve their writing and the layout of the paper.

---

> ### Author Response · Authors · 2018-11-21
> **Response**
>
> We appreciate the Reviewer for taking the time to provide these comments. In response to these comments, we added additional experiments and details to our revision.
>
> Our paper introduces 1) the correction module that directly updates predictions 2) input of a representation of all the training data T_S^s and 3) training of the correction module using an episodic procedure.
>
> Comment: The presentation of this paper is very poor.
>
> We improved the presentation, layout and writing of this paper. We expanded upon additional details in new appendices.
>
> Comment: experiments are not persuasive enough to demonstrate the significance of the proposed framework
>
> The experiments demonstrate that our model achieves state of the art accuracies, with a relative improvement of up to 10% in the conventional zero-shot learning setting. We added experiments in the generalized zero-shot learning setting. Our model consistently outperforms the runner up, achieving relative improvements of up to 31%.
>
> Comment: The proposed zero-shot classifier is based on the nearest centroid...
>
> We found a single centroid to work well. In these benchmark datasets, there are a lot of classes but not many samples per class. In these datasets, the size of the dataset per class is too small.
>
> Comment: The classification performance could be poor if two the mean of different classes close to each other [sic]
>
> If the feature extractor performs poorly, we can add regularization to increase the distance between means. We used a pre-trained feature extractor and found that they were already well divided.
>
> Comment: - Another confusion I have is why the training of the task module is not based on a fixed correction module?
>
> The task module is trained independently of the correction module. Thus, the task module is not based on a correction module. If the task module was trained with the correction module, then the task module would be indirectly training on unseen classes. This would defeat the purpose of the correction module learning to update predictions on unseen classes, because the unseen classes would no longer be unseen with respect to the task module. There is only one correction module that is trained across different episodes of the task module, where in an episode, the task module is trained on a sampled subset of seen classes.
>
> Comment: - The experiments also have many problems.  Authors need to clearly state how they construct meta-training, validation and testing instance.
>
> The testing splits are from published splits from Elhoseiny et al., 2017b. To create validation sets, we randomly divided the published training data by class into a validation and a “training” split. The validation set was created by randomly sampling classes from the published training set to create a validation set of the same size as the published training set. For example, CUB has 200 classes with the conventional SCS split having 50 test classes. We divided the remaining 150 classes into a 50 class validation set and a 100 class “training” set. When a validation set is used, the “training” split is treated as the seen data S. To report the test numbers, the original published training data is treated as the seen data S. We describe this further in the new Appendix A. Training of the task module (learner) and the correction module (meta-learning) proceeds as in Algorithm 1.
>
> Comment: Since the proposed framework is a meta-framework, authors need to report their performance in different meta-train/test splits.  The conventional split of CUB and NAB is only considered as a single split.  How well the proposed framework generalizes to other meta splits?     How well the proposed method performance to a generalized zero-shot setting?
>
> To compare with previous papers, we need to use the same splits. We report numbers for 1) the SCS split and 2) the SCE split. The SCS split is the conventional split, with 150 training classes and 50 test classes for CUB. The SCE split is a different split with 160 training classes and 40 test classes for CUB. The SCS and SCE splits have published training and testing splits and performance numbers in previously published papers.
>
> We added experiments for the generalized zero-shot setting. Our model outperforms relative to the previous state of the art by up to 31% in the generalized zero-shot learning setting.
>
> Comment: There are many typos. Auhtors definitely need to improve their writing and the layout of the paper [sic]
>
> We corrected the writing typos and improved the layout of the paper in our revision.

---

### Official Review · AnonReviewer3 · 2018-11-01

**Rating:** 4
**Confidence:** 5

**Review:**

Summary: This paper proposes a “meta-learning” approach for zero-shot learning. There is a Task Module that works in a conventional zero-shot way: Training to predict a class prototype using the auxiliary/text data description of that task. The new part is the added Correction Module that inputs both the target/zero-shot task description, the training task description, and the current prediction of the task module, and then outputs a correction vector that is added to the output of the task-module to produce the final output. The resulting system achieves state of the art results on zero-shot fine-grained classification (CUB and NAB).

Assessment: Overall this might be a good idea worthy of publication at some point. But despite the good results, the current realisation is not well analysed about exactly how and why it works, with no insight being provided; and leaves some doubt about the validity of the comparative experiments. The writing is also very rushed. It is not ICLR standard yet.

Strengths:
+ Interesting idea overall.
+ Good results.
Weaknesses:
- Poor clarity.
- Some experimental evaluation questions.
- Poor analysis.

Comments:
1. The correction module inputs the full set of training features T_s (Alg1-L13). However the training dataset is fixed, therefore this input is effectively a constant. So its not clear how a constant input can possibly be useful.
1.1 Possibly this has something to do with the episodic training, but this is exactly the kind of thing that should be analysed and explained, but is not discussed at all.
2. The paper is sold as a meta-learning paper, but it’s not clearly explained what is the “meta” part of the algorithm.
3. Its not explained anywhere how exactly the T_s, T_s^u, etc are fed into the correction network. Is it average pooling? It seems that simple average pooling is unlikely to be adequate given the large number (150) of classes in CUB.
4. There are no experimental details such as hyper parameters, network architecture, etc.
5. Based on the ablation study (Tab 2), the baseline task network without correction network already achieves state of the art results. Conceptually the task-network alone is a very standard “regression” based approach to ZSL of the type that people tried almost 10 years ago. So what is the explanation for why its so good? This makes the comparison to all the competitors in Tab1 suspect. If there is some reason (E.g., better image feature extractor or pre/post-processing) that makes the ultra simple baseline there already outperform SotA, then you have to ask how all the prior methods would perform if they were run with the same tweak.
6. Overall no insight provided about what kind of corrections are made, when they are useful, etc. This is important to provide insight about how/why correcting outputs can work.
7. There is nothing particularly unique about this setup for ZSL. It could equally be applied to correct outputs in the case of few-shot learning (CF: Prototype Networks). It would be more convincing if it was applied to both settings and analysed better for both.
8. The writing is very rushed. There are lots of writing and editorial errors. To name a few: P4 Extra “Task module is trained to minimise.” P4 “\mu_u” Is repeated. Citation style “Mohamed Elhoseiny & Elgammal” is wrong, check the bibtex.

---

> ### Author Response · Authors · 2018-11-21
> **Response to Reviewer's comments**
>
> We thank the reviewer for reading our paper and providing these comments. We answered the reviewer’s questions below and integrated the answers into our revised draft.
>
> Comment: 1. The correction module inputs the full set of training features T_s (Alg1-L13). However the training dataset is fixed, therefore this input is effectively a constant. So its not clear how a constant input can possibly be useful.
>
> The set of training features is not constant. The notation of T_s (Alg1-L13) has been updated to T_s^S. In the episodic training approach, the task module is trained on a random subset of all available training classes. The correction module takes as input the set of training features used to train the task module. Thus, the set of training features is not fixed because the task module’s training data is randomized every episode.
>
> Comment: 2. The paper is sold as a meta-learning paper, but it’s not clearly explained what is the “meta” part of the algorithm.
>
> The ‘meta’ part of the algorithm refers to the episodic training approach where in each episode, a ‘learner’ is trained from subsampled classes, while across episodes, a ‘meta-learner’ is trained using the ‘learners’. The reviewer observed that the Task Module works in a conventional zero-shot way. As such, the task module is the ‘learner'. The correction module is the ‘meta-learner’. The correction module is trained across different episodes of the task module to improve the prediction.
>
> Comment: 3. Its not explained anywhere how exactly the T_s, T_s^u, etc are fed into the correction network. Is it average pooling?
>
> The T_s (or T_s^u or T_s^u) is fed into linear feedforward layers followed by sum pooling across the number of classes. Additional details are provided in the new Appendix A. We found that simple average pooling did not perform as well.
>
> Comment: 4. There are no experimental details such as hyper parameters, network architecture, etc.
>
> Experimental details such as hyperparameters, network architectures, etc. are elaborated upon in the new additional Appendix A.
>
> Comment: 5. Based on the ablation study (Tab 2), the baseline task network without correction network already achieves state of the art results. Conceptually the task-network alone is a very standard “regression” based approach to ZSL of the type that people tried almost 10 years ago. So what is the explanation for why its so good? This makes the comparison to all the competitors in Tab1 suspect. If there is some reason (E.g., better image feature extractor or pre/post-processing) that makes the ultra simple baseline there already outperform SotA, then you have to ask how all the prior methods would perform if they were run with the same tweak.
>
> It is expected that our baseline task module only network performs at or slightly better than SotA (Zhu et al, 2018) because we use the exact same published input features and we use Zhu et al.’s architecture trained as a tuned feedforward network. Thus, it is expected that our task-module alone would perform similarly to Zhu et al. We compare to the exact same competitors as Zhu et al. When new architectures, image features extractor, or pre/post processing procedures are introduced for this problem, they can be used for the task module in our model.
>
> Comment: 6. Overall no insight provided about what kind of corrections are made, when they are useful, etc. This is important to provide insight about how/why correcting outputs can work.
>
> This is an interesting topic and we will think about this in future work. The corrections in our experiments are additive to the initial prediction. An idea is to use an interpretable representation of the image embedding. Then, additive corrections can be viewed as interpretable deviations in this embedding space.
>
> Comment: 7. There is nothing particularly unique about this setup for ZSL. It could equally be applied to correct outputs in the case of few-shot learning (CF: Prototype Networks). It would be more convincing if it was applied to both settings and analysed better for both.
>
> It is an interesting idea to extend our model to correct outputs for few-shot learning. We added this to our future work section.
>
> Comment: 8. The writing is very rushed. There are lots of writing and editorial errors. To name a few: P4 Extra “Task module is trained to minimise.” P4 “\mu_u” Is repeated. Citation style “Mohamed Elhoseiny & Elgammal” is wrong, check the bibtex.
>
> We thank the reviewer for pointing out the writing and editorial errors. We have edited and revised the writing.

---

### Meta-Review · Area_Chair1 · 2018-12-14
**Novel approach, but needs stronger comparisons.**

**Confidence:** 4
**Recommendation:** Reject

**Metareview:**

This is a difficult decision, as the reviewers are quite polarized on this paper, and did not come to a consensus through discussion. The positive elements of the paper are that the method itself is a novel and interesting approach, and that the performance is clearly state of the art. While impressive, the fact that a relatively simple task module trained on the features from Zhu et al. can match the performance of GAZSL suggests that it is difficult to compare these methods in an apples-to-apples way without using consistent features. There are two ways to deal with this: train the baseline methods using the features of Zhu, or train correction networks using less powerful features from other baselines.

Reviewer 3 pointed this out, and asked for such a comparison. The defense given by the authors is that they use the same features as the current SOTA baselines, and therefore their comparison is sound. I agree to an extent, however it should be relatively simple to either elevate other baselines, or compare correction networks with different features. Otherwise, most of the rows in Table 1 should be ignored. Running correction networks in different features in an ablation study would also demonstrate that the gains are consistent.

I think the authors should run these experiments, and if the results hold then there will be no doubt in my mind that this will be a worthy contribution. However, in their absence, I can’t say with certainty how effective the proposed method really is.

---

> ### Author Response · Authors · 2019-01-07
> **Methods were compared using consistent features - Reported baseline methods already use the features from Zhu**
>
> We appreciate the reviewer's description of the details of their decision.
>
> The authors of Zhu et al. confirmed that the baseline methods in Zhu et al. that we compared to were trained using the same features of Zhu et al. These experiments were done in Elhoseiny et al, CVPR 2017.  We were able to reach the authors of Zhu et al and Elhoseiny et al and verified this.
>
> Specifically, the same semantic and visual representations were used for GAZSL, ZSLPP, ESZSL, WAC-linear, WAC-kernel, and ZSLNS (and also correction networks). Thus, the methods are compared using consistent features.